# Relationship between Physical Activity and Sedentary Behavior, Spinal Curvatures, Endurance and Balance of the Trunk Muscles-Extended Physical Health Analysis in Young Adults

**DOI:** 10.3390/ijerph20206938

**Published:** 2023-10-18

**Authors:** Verner Marijančić, Tanja Grubić Kezele, Stanislav Peharec, Nataša Dragaš-Zubalj, Sandra Pavičić Žeželj, Gordana Starčević-Klasan

**Affiliations:** 1Department of Physiotherapy, Faculty of Health Studies, University of Rijeka, 51000 Rijeka, Croatia; verner.marijancic@uniri.hr (V.M.); stanislav.peharec@fzsri.uniri.hr (S.P.); 2Department of Physiology, Immunology and Pathophysiology, Faculty of Medicine, University of Rijeka, 51000 Rijeka, Croatia; 3Department of Clinical Microbiology, Clinical Hospital Rijeka, 51000 Rijeka, Croatia; 4Department of School and University Medicine, Teaching Institute of Public Health of Primorje-Gorski Kotar County, 51000 Rijeka, Croatia; natasa.dragas-zubalj@zzjzpgz.hr; 5Department of Health Ecology, Teaching Institute of Public Health of Primorje-Gorski Kotar County, 51000 Rijeka, Croatia; sandrapz@medri.uniri.hr; 6Department of Basic Medical Science, Faculty of Health Studies, University of Rijeka, 51000 Rijeka, Croatia; gordanask@fzsri.uniri.hr

**Keywords:** physical activity, physical fitness, posture, quality of life, sedentary lifestyle, sleep quality, spinal curvatures, trunk muscle endurance, young adults

## Abstract

Background: Physical inactivity and sedentary behavior are associated with poor well-being in young people with adverse effects extending into adulthood. To date, there are many studies investigating the relationship between physical activity (PA) and posture, but there are no data on the relationship between the type and intensity of PA and sedentary behavior, their association with thoracic and lumbar spine angles, and with endurance and balance of the trunk muscles, especially in healthy young adults aged 18–25 years. Moreover, there are no data on the relationship between PA and sedentary behavior and musculoskeletal and cardiopulmonary health, as well as quality of life (QoL) and sleep that would provide a more comprehensive picture of physical health status. Aim: Therefore, the aim of this cross-sectional study was to investigate the extent to which PA and sedentary behavior are associated with each other and with changes in spinal curvatures, endurance and balance of trunk muscles in an extended analysis of physical health status in young adults aged 18–25 years by additionally including measures of body composition, cardiorespiratory capacity, and QoL and sleep. Methods: A total of 82 students (58% female, 42% male) aged 18–25 years completed all required tests. Primary outcome measures included the following: PA and sedentary behavior calculated from the long form of International PA Questionnaire (IPAQ-LF), spinal curvatures measured by a Spinal Mouse^®^ device, endurance and balance of the trunk muscles measured using trunk endurance tests and their ratio. Results: Overall, 50% of students were classified as minimally active and 50% as health-enhancing PA (HEPA) active. The angles of thoracic kyphosis and lumbar lordosis showed no correlation with PA or time spent sitting. However, students with the lowest PA had significantly higher scores on the trunk extensor endurance test and trunk extensor/flexor endurance test ratio, indicating imbalanced trunk muscles. Moreover, these students spent the most their time sitting. Only PA of vigorous intensity and PA during recreation, leisure, and sports significantly correlated with QoL related to physical health. QoL related to physical and psychosocial health had significantly higher scores when students spent less time sitting. In addition, we found significantly better respiratory performance and SQ at higher PA values, i.e., PA during recreation, leisure, and sport. Conclusions: Our results suggest that students with low PA levels and more time spent sitting have imbalanced trunk muscles, worse respiratory function, and poorer QoL and sleep. Moreover, these findings in college students may reflect their lifestyle and suggest that more PA needs to be promoted to prevent the development of chronic diseases including musculoskeletal disorders.

## 1. Introduction

Physical inactivity, poor physical condition, and sedentary behavior among young people are a growing public health problem worldwide [1,2]. According to the World Health Organization (WHO), physical inactivity and sedentary behavior are the 4th leading cause of death in the world and are estimated to be the cause of about 6–10% of deaths in the non-communicable disease (NCD) group, such as chronic diseases, i.e., obesity, cardiovascular diseases, hyperlipidemia, hypertension, type 2 diabetes, metabolic syndrome, and cancer [3,4]. The current WHO recommendations of the WHO’s Global Action Plan on Physical Activity 2018–2030 (GAPPA) for adults aged 18 to 64 years are as follows: at least 150 min of moderate and 75 min of vigorous physical activity (PA) per week (wk) and limiting time spent sitting [5]. Yet today, more than 80% of adolescents and 27% of adults do not achieve the WHO’s recommended levels of physical activity [4,5]. This not only affects individuals throughout their lives, but also places a financial burden on health services and society as a whole. In addition, epidemiological studies show that too much sedentary behavior not only increases the risk of chronic diseases [5,6,7,8,9], but also suggests a link between sedentary behavior and all these diseases, regardless of PA level [10].

College students, i.e., young adults, are spending increasing amounts of time sitting while studying, watching TV, or playing computer and console games, and these risky behaviors pose a threat to their future health [11], making them a particularly vulnerable population [12]. Recent studies have shown the importance of assessing the quality of life (QoL) of young adults, i.e., college students, a population in transition to adulthood, as this phase of life can be an important time to monitor and intervene in lifestyle habits that have a major impact on future life practices [13,14,15,16,17,18]. In addition to the known health risks of reduced PA, a sedentary lifestyle is also thought to negatively impact musculoskeletal health and lead to postural changes in young adults (Appendix A) [10,16,17,18,19,20]. Good posture is defined as the state of balance between muscles and skeletal segments that is essential for maintaining postural balance in both static and dynamic positions of the body [21]. Thus, imbalance of the trunk extensor and trunk flexor muscles because of the poor posture can alter the curvature of the lumbar spine, increase lordotic curvature, and cause health problems later in life [22]. In addition, the predominant sedentary lifestyle not only shortens or weakens the back muscles [23], but physical inactivity also leads to joint contractures and narrows the spinal joints [24]. Sedentary behavior also causes changes in muscle fibers that may contribute to muscle stiffness, especially in the lumbar extensors, the muscles required for an upright posture when sitting [25,26].

Furthermore, poor posture during the college period can lead to chronic health issues, back pain [26,27,28], decreased respiratory function due to reduced lung capacity [29], decreased core strength and stability, stress [21,22,23,24], poor self-esteem and subsequently lower QoL (Appendix A) [13,14,15,16,17,18]. Additionally, studies have shown the negative impact of physical inactivity and sedentary behavior on students’ QoL caused by poor sleep quality (SQ) [17,30]. If SQ has deteriorated because of the consequences of poor posture, it means that physical changes are already present in the spine [21]. SQ can, however, be directly improved by regular PA [21]. Nevertheless, regular PA not only leads to positive changes in the human body that make it possible to prevent all these above-mentioned disorders [8,9], but can also improve posture and, subsequently, QoL. In addition, it is already known that QoL can be directly improved by PA increasing physical fitness (Appendix A) [7].

Furthermore, physical fitness is an important determinant of health and is related to the musculoskeletal system and the overall biological functioning of the body [31]. This is because PA not only improves strength and muscular endurance, but also the efficiency of respiratory processes and the rate of blood flow, leading to a more effective supply of oxygen and nutrients to organs and tissues, including the brain, which in turn increases a sense of satisfaction (Appendix A) [31].

In the relevant literature, there are many research reports worldwide investigating the relationship between PA and BMI, body composition, physical fitness, QoL, and posture in healthy populations, including university students [13,14,15,16,17,18,19,20,21,22,23,24,25,26,27,28,29,30,31,32,33,34,35,36]. However, the association between PA and sagittal posture is controversial. There is evidence that more physically active participants have better postural control compared to inactive subjects [32]. On the contrary, a study on lumbo-pelvic sagittal posture showed that there was no association between lumbo-pelvic posture and the level of PA [33]. Furthermore, a study on the relationship between prolonged sedentary behavior and thoracic spine mobility showed that the thoracic spine is less mobile in participants who sit for more than 7 h per day and are physically active for less than 150 min (min) per week [34]. Despite an increasing volume of research focusing on college students, i.e., young adults aged 18–25 years, there remain major gaps in our knowledge about how type and intensity of PA and furthermore, level of sedentary behavior, are associated with each other, and with changes in posture, i.e., with changes in endurance and balance of the trunk muscles, and physiological spinal curvatures. In addition, to our knowledge, there is not yet a study investigating the relationship between PA and sedentary behavior and musculoskeletal and cardiopulmonary health, as well as QoL and sleep, to provide a more comprehensive picture of the physical health status of young adults.

Therefore, the purpose of this cross-sectional study was to investigate the extent to which PA and sedentary behavior are associated with each other and with changes in physiological spinal curvatures and endurance and balance of the trunk muscles in an extended analysis of the physical health status of young adults aged 18–25 years by additionally including measures of body composition, cardiorespiratory capacity, and QoL and sleep. We hypothesized that students with low PA levels and more time spent sitting would have increased trunk extensor endurance and imbalanced trunk muscles, poorer physical fitness, and poorer quality of life and sleep.

## 2. Materials and Methods

### 2.1. Participants

The present study was a cross-sectional study conducted at the University of Rijeka, Croatia during the 2022–2023 academic school year. Participants were recruited from three different faculties: Faculty of Health Studies, Medicine, and Maritime Studies. 

An estimation of the appropriate sample size for the study was guided by these two methods: (1) previous similar research and (2) general statistical principles; more specifically: (1)Sample sizes in similar research ranged from 77 [35] to 168 [36].(2)The program MedCalc (© 2023 MedCalc Software Ltd., Ostend, Belgium) estimated a minimum of 123 subjects was needed to achieve 80% power with r = 0.25, α = 0.05 type I error and beta = 0.20 type II error.

After being introduced to the research project through a public presentation, an interview was conducted with 145 volunteers. After an interview, a total of 126 volunteers met the inclusion criteria and were enrolled in the study. The inclusion criteria were healthy university students aged between 18 and 25, i.e., young adults, with no cardiovascular, respiratory, metabolic, autoimmune, or other systemic diseases and/or spinal pathologies, no previous diagnosis of a systemic musculoskeletal problem or pain, and no history of spinal or extremity surgery. Individuals who used assistive devices or orthoses were also excluded from the study. The researchers obtained the necessary approvals from the Institutional Ethics Committee of Teaching Institute of Public Health (number: 08-820-40/50-22) and Ethics Committee of the Faculty of Health Studies at the University of Rijeka (number: 2170-1-65-23-1). In accordance with the Declaration of Helsinki, informed consent was obtained from all participants. 

### 2.2. Study Design

The cross-sectional study design followed the STROBE Statement [37]. All measurements were performed by the same researchers. A research protocol designed for this study consisted of three phases. The 1st phase consisted of filling out the questionnaires. The 2nd phase consisted of 2 subphases; in the 1st subphase, the Harvard Step Test (HST), body composition analysis, and spirometry were performed, and in the 2nd, endurance tests were performed. The 3rd and final phase was the measurement of spinal curvatures. There was a 24-h break between the three main phases, and 24 h between the HST and the endurance tests.

### 2.3. Outcome Measures

#### 2.3.1. Questionnaires

##### Self-Reported Quality of Life

QoL was measured using the Pediatric Quality of Life Inventory 4.0™ (PedsQL 4.0™) self-report for young adults (age 18–25 years) (Mapi Research Trust, Lyon, France, https://eprovide.mapi-trust.org; accessed on 13 August 2022.) [38,39]. It is a reliable and valid 23-item questionnaire composed of the following 4 dimensions: (1) physical functioning (8 items); (2) emotional functioning (5 items); (3) social functioning (5 items); and (4) work/school functioning (5 items). The reliability values (Cronbach’s alpha) range from 0.71 to 0.86 [38], and validity values (effect sizes according to Cohen) range from 0.27 to 1.11 [40]. Scale scores are computed as the sum of the items divided by the number of items answered.

##### Self-Reported Sleep Quality

SQ was measured using the Pittsburgh Sleep Quality Index (PSQI). The PSQI is a reliable and valid self-rated questionnaire that assesses SQ and disturbances over a 1-month time interval [41,42]. The seven component scores of the PSQI have an overall reliability coefficient (Cronbach’s alpha) of 0.83 [41]. The content validity index of PSQI is 0.91 [43]. The test distinguishes poor quality from good quality sleep by measuring seven components that include: subjective SQ, sleep duration, sleep disorders, use of sleep medications, and daily disorders that have occurred in the past month. A score ≥ 5 represents poor sleepers and score < 5 represents individuals with normal SQ.

##### Self-Reported PA and Time Spent Sitting

The IPAQ-long form (IPAQ-LF) was administered by trained interviewers to assess participants’ self-report PA and sedentary behavior [44,45]. It is a reliable and valid questionnaire that health education and promotion professionals can confidently use to assess college students’ participation in PA [46]. The validity indices of the questionnaire are similar to other self-report PA questionnaires [47]. A detailed description of the IPAQ assessment protocol, including the criteria for cutting off extreme values, is available online [44]. According to WHO recommendations [3], minimal PA per week was considered ≥150 min/wk of moderate or ≥75 min/wk of vigorous PA (sufficient PA per week) and values of weekly PA below the minimum (insufficient PA per week) was considered <150 min/wk of moderate or <75 min/wk of vigorous PA. According to IPAQ-LF categorical scoring, inactive PA level was the lowest level of PA with less than 600 MET-min/wk, minimally active PA level was a minimal total PA with at least 600 MET-min/wk, and health-enhancing PA (HEPA) active level was a minimal total PA of at least 1500 MET-min/wk [44].

#### 2.3.2. Body Composition Analysis and Body Mass Index

Analyses of participants’ body composition were performed using the Tanita RD-545 Ma Segmental Body Composition Analyzer (Tanita, Japan). The Tanita RD-545 is a reliable and valid single-frequency bio-impedance analyzer with 8 polar electrodes [48,49]. In this study, visceral fat, body fat percentage, muscle percentage, and physique rating were determined. Ratings from 1 to 12 were considered a healthy level of visceral fat for females and males, while ratings from 13 to 59 were considered excessive [48]. The percentage of body fat for females was determined as follows: underfat (0–20.9%), healthy (21–32.9%), overweight (33–38.9%), and obese (≥39%); and for males as follows: underfat (0–7.9%), healthy (8–18.9%), overweight (19–24.9%), and obese (≥25%) [48]. The percentage of muscle mass was calculated as muscle mass (kg)/total body weight (kg) × 100 = muscle mass (%). The percentage for muscle mass for females was determined as follows: very low (<56%), low (56–61%), good (62–75%), and increased (>75%); and for males as follows: very low (<71%), low (71–75%), good (76–88%), and increased (>88%) [48]. Physique ratings were determined as follows: 1-Hidden Excess Fat, 2-Medium Frame & Excess Fat, 3-Solidly Built, 4-Low Muscle, 5-Standard, 6-Muscular, 7-Low Muscle & Low Fat, 8-Thin & Muscular (Athlete), and 9-Very Muscular (Athlete) [48].

Body Mass Index (BMI) was calculated as a participant’s weight in kilograms divided by the square of height in meters (kg/m^2^). Underweight was considered <18.5 kg/m^2^, normal between 18.5 and 24.9 kg/m^2^, overweight between 25 and 29.9 kg/m^2^, obese between 30–34.9 kg/m^2^, and extremely obese >35 kg/m^2^ [48].

#### 2.3.3. Trunk Muscles Endurance Testing

##### Trunk Extensor Endurance Testing

The trunk extensor endurance test is a reliable and valid test for assessing the muscular endurance of the torso extensor muscles that stabilize the spine (i.e., erector spinae and multifidus muscles) [50,51,52]. This is a timed test with a static, isometric contraction performed according to the modification of McGill et al. [50]. Participants were instructed to lie prone on a test table. The trunk was at the level of the anterior superior iliac spine at the edge of the test table. Participants kept their upper body away from the end of the table by supporting themselves with their outstretched arms on a chair directly below them. The test time was set at 180 sec and measured with a stopwatch while the arms were lifted from the chair and crossed over the chest with the hands resting on the opposite shoulders and the participants assuming the horizontal position. The test was terminated when participants visually deviated from the horizontal plane. The test time was noted. Clinician 1 measured the time participants and stood sideways and used the aforementioned criterion of participant deviation from the horizontal plane as the criterion for termination of the extensor endurance test. Clinician 2 stabilized the participants’ lower bodies by holding the participants’ lower extremities down [52].

##### Trunk Flexor Endurance Testing

A standard trunk flexor endurance test was performed according to previously published methods [50]. The trunk flexor endurance test is a reliable and valid test that evaluates the muscular endurance of the trunk flexors (i.e., rectus abdominis, external obliques, internal obliques, and transversus abdominis muscles) [51,52]. This is a timed test in which the anterior muscles are isometrically contracted to stabilize the spine until the subject shows signs of fatigue and can no longer maintain the assumed position or reaches the specified time of 180 s. Participants were in the supine position with hips and knees flexed to 90° and the trunk inclined at 60° resting on a wedge. Arms were crossed in front of the chest and hands were placed on opposite shoulders. Time was measured from the moment the wedge was pushed back 10 cm until the participant visually reestablished contact with the wedge. Clinician 1 stood at the participant’s side and used the above criterion of the participant’s visual re-contact with the wedge as the criterion for completion of the flexor endurance test. The time was noted. Stabilization of the participant’s feet was performed by clinician 2 [52].

##### Balance of Trunk Muscles 

The trunk extensor/flexor endurance test ratio represents a good parameter of trunk muscle balance. It is a ratio between the endurance of the trunk extensor muscles and the endurance of the trunk flexor muscles. This measure is calculated from the ratio between the trunk extensor endurance and trunk flexor endurance scores. There are no reference values for this ratio. It was modified following Kim et al. [53].

#### 2.3.4. Cardiovascular Fitness 

The Harvard Step Test is a reliable and valid test used to assess cardiovascular fitness or Physical Fitness Index (PFI) [54]. The instruments for the HST were two boxes, 50.8 cm high for male participants and 40 cm high for female participants, a stopwatch, and a metronome. The step test was recorded at the rhythm of the metronome (120 Hz) as 1 cycle in 2 s, consisting of the following: 1st foot up on the 1st beat of the metronome, 2nd foot up on the 2nd beat, 1st foot down on the 3rd beat, 2nd foot down on the 4th beat. Participants were instructed to perform the exercise at the same speed. Time was measured with a stopwatch. The test was stopped after 5 min or until exhaustion. Exhaustion is defined as the point when participants can no longer maintain the stride rate for 15 s. The time was noted. After completion of the test, pulse was measured in the seated position 1 to 1.5 min after the test, 2 to 2.5 min after the test, and 3 to 3.5 min after the test. The PFI was calculated using the formula below, where t_e_ is the exhaustion time and t_b_ is the total number of heartbeats:(1)$$PFI=\frac{t_{e}\times 100}{t_{b}\times 2}$$

After calculating the PFI for each participant, scores were assigned as follows: <55 (poor physical condition), 55–64 (below average), 65–79 (average), 80–89 (good), and ≥90 (excellent) [54]. Blood pressure and heart rate were measured before HST.

#### 2.3.5. Spirometry

Spirometry is performed with a reliable SPIROLAB^®^ spirometer (Rome, Italy) to measure maximal voluntary ventilation (MVV), which correlates with respiratory muscle performance [29,55,56,57]. MVV measurement was used to estimate the overall capacity of the respiratory system to move air, respiratory muscle strength, compliance of the pulmonary-thoracic system, and airway resistance [55]. The MVV is the largest volume that can be inhaled and exhaled into the lungs during a 12-s interval with maximal voluntary effort [55]. The MVV test does not last a full minute because of the extreme pH changes in the body as well as dizziness and fainting in subjects.

In this study, subjects wore nose clips and breathed deeply (with a volume greater than tidal volume but less than vital capacity) and rapidly for a 12-s interval. After discarding the first three to five breaths, subjects were actively encouraged to maintain the same volume and rate by following an online display of the maneuver on a computer screen, i.e., the end-expiratory level remained relatively constant [55]. At least two acceptable maneuvers (with no more than a 10% difference between them) were performed, and, after flow integration, the highest value was determined by extrapolating the 12-s accumulated volume to 1 min (L/min). A normal MVV value for healthy college-aged men ranges from 140 to 180 L/min [56,57]. The range for healthy college-aged women is 80–120 L/min. In general, trained individuals have better MVV performance than untrained individuals [29].

#### 2.3.6. Evaluation of Spinal Curvatures

Spinal curvatures, i.e., the angle of thoracic kyphosis and lumbar lordosis, were measured using a noninvasive Spinal Mouse^®^ (SM) (Idiag M360, Fehraltorf, Switzerland) device. It is a safe, reliable, quick, and easy-to-use method with no side effects, and a proper substitute for radiography images for measuring angle values of thoracic kyphosis and lumbar lordosis [58,59,60,61]. Posture measurements were performed in the sagittal plane with bare feet in relaxed standing or anatomical positions using the Idiag M360 protocol software version G6 6.4 2X. The measurements were performed on one day, and no exercises were performed before the measurement. Using the software of this device, the data displayed on the screen were used to study the positional relationship between each vertebra, measure the angles between vertebrae, and calculate the angles of spinal curvatures. The standard procedure for the upright sagittal posture was performed. The spinous process of the 7th cervical vertebra was marked as the starting point for the measurement and the end point was marked at the level of the 3rd sacral vertebra. The posterior superior iliac spine (PSIS) was marked using an alternative method by drawing the line between PSIS. After the line between PSIS was drawn 2 cm below the line, the position was marked with a flexible ruler. The vertical line was used to mark the center of the new line, below the PSIS line, so that the cross was above the S3 vertebra. The SM is placed over the C7 vertebra with the orange mark on the device above the marked starting position and recording is done by moving the device from top to bottom to the end point. Values of thorax T1/T2 to T11/12 and lumbar spine T12/L1 and L1 to S1 were recorded. Negative values in the lumbar curve correspond to lumbar lordosis. In the assessment of standing position, values between 20° and 45° were considered neutral thoracic kyphosis, less than 20° was considered hypokyphosis, and more than 45° was considered hyperkyphosis [58]. The values of the lumbar spine in relation to lordosis were considered for neutral lordosis ranging from 20° to 40°. The values less than 20° were classified as hypolordosis and an angle more than 40° as hyperlordosis [62].

### 2.4. Statistical Analysis

Data were analyzed using Statistica, version 13 (TIBCO Software Inc, 2017, Palo Alto, CA, USA). Because there are no established thresholds for presenting MET-minutes (dependent variables), the IPAQ Research Committee proposes that these data be presented as comparisons of median values and interquartile ranges for different populations [44]. Participants were divided into quartiles of PA levels using IPAQ-LF: 672–2087, 2088–4008, 4009–8573, and 8574–22,062 MET-min/wk (dependent variables). Throughout the text, the following symbols for PA quartiles are used: “Q1”, “Q2”, “Q3” and “Q4”. The mean values, standard deviations, standard errors, medians, ranges, and percentages of the descriptive data were calculated. Data distribution was tested for normality using the Kolmogorov–Smirnov test. 

The IPAQ domains were given as mean values ± standard error (SE).

The percentages of participants in the PA quartile groups were calculated by sex, BMI (underweight, normal, overweight, obese, extremely obese), visceral fat level (healthy percentage or excess level rating), body fat percentage (underfat, healthy, overweight, obese), muscle mass (very low, low, good, increased), minimum required PA per week (<150 min/wk of moderate or <75 min/wk of vigorous PA were considered as insufficient PA per week; ≥150 min/wk of moderate or ≥75 min/wk of vigorous PA were considered as sufficient PA per week), PA level (inactive, minimally active, health-enhancing physical activity (HEPA) active), SQ (good and poor), cardiovascular fitness (poor, low average, average, good, excellent), MVV (males: 140–180 L/min; females: 80–120 L/min) and separately, spinal curvatures (kyphosis and lordosis angles). 

Chi-square analyses were used to examine the frequency distributions of all these independent variables between the PA quartiles. For BMI, since there were no students in “underweight” and “extremely obese” categories, the chi-square analysis included only “normal”, “overweight”, and “obese” variables. For visceral fat level, only a healthy level of visceral fat was found, and therefore was presented as number of students and percentage only. For PA level, since there were no “inactive” students, the chi-square analysis included only “minimally active” and “HEPA active” variables. For thoracic kyphosis angles, since there were no hypokyphotic students, the chi-square analysis included only “neutral kyphosis” and “hyperkyphosis” variables.

The independent variables BMI, body composition (visceral fat level, body fat, muscle mass, physique rating), PA (PA in MET-min/wk, time spent sitting in min/wk, PA level), quality of life (physical, psychosocial and total), SQ, trunk muscle endurance (trunk flexor and extensor endurance in sec, trunk extensor/flexor test ratio), spinal curvatures (kyphosis and lordosis angles), cardiovascular fitness (poor, below average, average, good, excellent), and MVV were presented as the mean values ± standard deviations (SD) in the PA quartile groups (dependent variables).

To compare the mean values of all these independent variables between the PA quartiles the one-way ANOVA post-hoc Tukey test was used for multiple parametric data and the Kruskal–Wallis test for multiple non-parametric data.

The relationships between the measurements were analyzed using Pearson correlation analyses. The following dependent variables were considered: vigorous PA (MET-min/wk), PA during recreation, sport and leisure time (MET-min/wk), total PA (MET-min/wk), and time spent sitting (min/wk). The following independent variables were considered: BMI, body composition (visceral fat level, body fat, muscle mass, physique rating), quality of life (physical, psychosocial and total), SQ, trunk muscle endurance (trunk flexor and extensor endurance in sec, trunk extensor/flexor test ratio), spinal curvatures (kyphosis and lordosis angles), cardiovascular fitness (poor, below average, average, good, excellent), and MVV. In the correlation analyses, the values of the correlation coefficients were considered as follows: 0.00–0.19 were considered as a “no relationship”, 0.20–0.39 as a “weak relationship”, 0.40–0.69 as a “mid-level relationship”, 0.70–0.89 as a “strong relationship”, 0.90–1.00 as a “very strong relationship”. The significance level of the statistical analyses was set at *p* < 0.05.

## 3. Results

At baseline, a total of 126 subjects were enrolled in the study, and after the exclusion of 44 subjects who did not complete all required assessments, the study was completed with 82 subjects (Figure 1). To capture the physical health status of each individual more comprehensively, we only included the 82 subjects who completed all measurements. Of these, 48 (58.5%) were female and 34 (41.5%) were male. The mean age of the participants was 21.5 ± 1.4 years.

The descriptive data for the different domains of the IPAQ questionnaire for all 82 subjects can be found in Table 1. The mean ± SE of total PA was 5573.9 ± 494.4 MET-min/wk and the mean of time spent sitting was 2489.7 ± 105.1 min/wk. The type of PA with the highest number of MET-min/wk is PA during recreation, leisure, and sports (mean ± SE = 2469.6 ± 297.7 MET-min/wk), and the most frequently performed intensity of PA was walking (mean ± SE = 2184.8 ± 222.6 MET-min/wk).

Table 2 shows the percentages of participants in the PA quartile groups by gender, BMI, self-reported PA, body composition, self-reported SQ, cardiovascular fitness, and MVV. The higher the quartile, the higher PA. 

As for BMI, there were no underweight or extremely obese students in the study. Most students had a normal BMI (69.5%), but 25.6% of them were overweight and 4.9% were obese. There were no underweight and no extremely obese students registered. Furthermore, all students had a healthy visceral fat level (100%) and most of them had a healthy percentage of body fat (68.4%), and good percentage of muscle mass (76.8%).

As expected, Chi-square analyses revealed a significant difference in the frequency distribution in PA in min/wk [χ^2^ (3, N = 82) = 51.19, *p* = 0.000] and in PA level between the PA quartiles [χ^2^ (3, N = 82) = 10.96, *p* = 0.012] (Table 2), indicating that these PA quartiles are a valuable way to distinguish the amount of PA generated by the self-reported IPAQ questionnaire in this study [44]. We did not register any inactive students, i.e., students with PA less than 600 MET-min/week. Moreover, half of the students belonged to the highly active HEPA category (N = 41; 50%), and most of them were in Q4 (N = 16; 19.5%).

Chi-square analyses revealed a significant difference in the frequency distribution of good or poor SQ between the PA quartiles assessed by the self-reported PSQI, [χ^2^ (3, N = 82) = 8.37, *p* = 0.038] (Table 2). Table 2 shows that most subjects in Q1 have poor SQ (N = 16; 19.5%). However, most subjects show poor SQ in general (N = 51; 62.2%).

Most of the college students showed poor cardiovascular fitness (N = 34; 41.7%) and 25 (30.0%) of them had average cardiovascular fitness (Table 2). Only 4 (5.4%) of the students had MVV values below normal (Table 2).

Table 3 shows the percentages of participants in the PA quartile groups by spinal curvatures, i.e., thoracic kyphosis angle and lumbar lordosis angle. Overall, there was an increased percentage of hyperkyphotic students (N = 49; 58.9%), but most students with neutral lordosis were in Q4 (N = 21; 25.0%), suggesting better posture in more physically active individuals. However, no significant difference was found in the frequency distribution of hypokyphosis, neutral kyphosis, hyperkyphosis, hypolordosis, neutral lordosis, and hyperlordosis between the PA quartiles (Table 3). There were no hypokyphotic students registered.

Table 4 shows the mean values of the following variables: BMI, body composition, self-reported PA, self-reported quality of life, endurance and balance of the trunk flexors and extensors, cardiovascular fitness, MVV, and physiological spinal curvatures. 

No statistically significant difference was found in mean values of BMI and body composition between quartiles, although body fat percentage was highest in Q1, i.e., among students who were least physically active.

As expected, post-hoc analyses revealed statistically significant differences between the mean values of PA (MET-min/wk and min/wk) and time spent sitting between the quartile with the highest PA (Q4) and the quartile with the lowest PA (Q1) (*p* = 0.046) (Table 4). Students in the quartile with the highest PA (Q4) spent the least amount of time sitting, which is not always the case [8]. In addition, analysis of the Kruskal–Wallis test revealed statistically significant differences in PA level between the PA quartiles (*p* = 0.027) (Table 4). The highest PA level (HEPA level) was found in students in the last two quartiles with the highest PA (Q3 and Q4) (Table 4), which was also expected.

Furthermore, as the PA quartiles increased, the scores of QoL related to both physical and psychosocial health (measured by the self-reported PedsQL^TM^) gradually increased across the PA quartiles, but no significant differences in QoL related to physical and psychosocial health were found between the PA quartiles (Table 4). This suggests that the more physically active the subjects are, the better their QoL is, which has already been confirmed in many other studies [17,24,25,26,27,28]. However, in our study, the highest scores of QoL related to physical health were found in students in Q3, which was statistically significant compared to those of Q1 (*p* = 0.021). As expected, the highest QoL scores related to psychosocial health were found in students in the quartile with the highest PA (Q4), but with no statistically significant difference compared to other PA quartiles (Table 4). The differences in the SQ scores almost reached statistical significance, with the worst sleep quality found in Q1, where the students were least physically active (Table 4).

As the PA quartiles increase, the trunk flexor endurance test values also gradually increase across quartiles but with no significant differences found in trunk flexor endurance test values between quartiles (Table 4). However, these results suggest that the more physically active the subjects are, the greater the endurance of the trunk flexors (rectus abdominis, external obliques, internal obliques, and transversus abdominis muscles). The mean value of the trunk extensor endurance test was highest in students in Q1 (176.2 ± 13.8) with a significant difference compared to that of Q2 (*p* = 0.039) (Table 4), indicating that the muscles that stabilize the spine (i.e., erector spinae and multifidus muscles) seem to have the greater endurance in subjects with the lowest PA. The trunk extensor/flexor endurance test ratio was also highest in students in Q1 with a significant difference compared to those of Q2 (*p* = 0.014) and Q3 (*p* = 0.031) (Table 4). This suggests that the less PA, the more imbalanced the trunk muscle endurance.

There was no statistically significant difference found in kyphosis and lordosis angle between quartiles. However, the lordosis angle was most excessive in Q1, i.e., among the least physically active students (Table 4).

As expected, as the PA quartiles increased, the physical fitness index (as a reference value for cardiovascular fitness) also gradually increased across quartiles, from the lowest to the highest, but no significant differences in the physical fitness index were found between the PA quartiles (Table 4).

As the PA quartiles increased, the MVV as a value of respiratory muscle performance also gradually increased across quartiles, but no significant differences in MVV values were found between the PA quartiles (Table 4).

Table 5 shows Pearson correlation analyses for PA and time spent sitting vs. BMI, body composition, QoL and sleep, endurance and balance of the trunk muscles, cardiovascular fitness, MVV, and spinal curvatures. 

No correlation was found between body fat and muscle mass, and PA of moderate and low intensity or other types of PA such as walking, PA during transportation, PA at work, PA at home or gardening. 

In addition, no correlation was found between body fat and muscle mass with total PA (MET-min/wk) and time spent sitting (Table 5). Furthermore, only the PA of vigorous intensity and PA during recreation, leisure, and sports were positively correlated with muscle mass and negatively correlated with body fat (Table 5). Namely, body fat was significantly negatively correlated with PA of vigorous intensity (r = −0.254, *p* = 0.021) and with PA during recreation, leisure, and sports (r = −0.278, *p* = 0.011), while muscle mass was significantly positively correlated with PA of vigorous intensity (r = 0.260, *p* = 0.018) and with PA during recreation, leisure, and sports (r = 0.287, *p* = 0.009) (Table 5). 

The QoL related to physical health correlated only with PA of vigorous intensity (r = 0.374, *p* = 0.001) and with PA during recreation, leisure, and sports (r = 0.416, *p* = 0.000), and with total PA (MET -min/wk) (r = 0.301, *p* = 0.006) (Table 5). No correlation was found between QoL in general and PA of moderate and low intensity or other types of PA such as walking, PA during transportation, PA at work, PA at home or gardening. In addition, QoL related to physical and psychosocial health was inversely correlated with time spent sitting (r = −0.246, *p* = 0.026; r = −0.247, *p* = 0.025) (Table 5.), suggesting that the more time subjects spend sitting, the more their quality of life suffers in terms of physical and psychosocial health. No correlation was found between QoL related to psychosocial health and any type or intensity of PA (Table 5). 

In addition, a Pearson correlation showed a statistically significant inverse correlation between the PSQI scores and PA during recreation, leisure, and sports (r = −0.243, *p* = 0.028) (Table 5), suggesting that sleep quality increases the more intensively students engage with PA. 

No correlation was found between trunk muscle endurance, trunk extensor/flexor test ratio, nor spinal curvature angles with the total PA (MET-min/wk), time spent sitting, or with any type of PA or PA intensity level (Table 5). 

Cardiovascular fitness showed no correlation with any type of PA or PA intensity level or with sitting duration, and MVV correlated significantly only with recreation, leisure, and sport (r = 0.341, *p* = 0.010) (Table 5).

Appendix A shows Pearson correlation analyses for spinal curvatures vs. BMI, body composition, QoL and sleep, trunk muscle endurance, cardiovascular fitness, and MVV. Although this was not the primary goal of this study, there were positive correlations found between lumbar lordosis and body fat (r = 0.277, *p* = 0.038), trunk extensor endurance (r = 0.264, *p* = 0.049), and trunk extensor/flexor trunk test ratio (r = 0.323, *p* = 0.015), and negative correlations with muscle mass (r = −0.284, *p* = 0.034) and MVV (r = −0.289, *p* = 0.030) (Appendix A). However, no correlations were found between lumbar lordosis and BMI, visceral fat level, SQ, QoL, flexor trunk test and cardiovascular fitness. In addition, no correlations were found between the angle of thoracic kyphosis and any of these variables (Appendix A).

## 4. Discussion

College students are sedentary during most of their school and leisure time (this is also typical for office workers [63]) and this puts them at constant risk for future chronic cardiovascular, metabolic, and musculoskeletal disorders. In addition, this population is neglected, but the transition from adolescence to adulthood may be an important period to monitor, especially for lifestyle modification initiatives to reduce the above health risks. In fact, increasing PA and reducing sedentary behavior are the top priorities of the WHO’s GAPPA 2018–2030 [5]. Being more physically active not only has significant health benefits, but also, societies that are more active can also generate additional returns, such as reduced fossil fuel consumption, cleaner air, and less congested, safer roads. These outcomes are linked to achieving the shared goals, political priorities, and ambitions of the Sustainable Development Agenda 2030. [64].

### 4.1. PA and Sedentary Behavior (Time Spent Sitting)

In our study, according to IPAQ-LF, out of 82 students, 65 (79.3%) had moderate PA of more than 150 min per week or vigorous PA of more than 75 min per week and thus met the WHO recommendations on PA for health [64]. Overall, 50% of students were classified as minimally active and 50% as engaged in health-enhancing PA (HEPA). Interestingly, PA during recreation, leisure, and sports, PA of vigorous intensity, and total PA were negatively correlated with time spent sitting with high statistical significance. This suggests that the more physically active the students were, the more sitting time decreased. In addition, sitting time did not correlate with walking, PA of moderate intensity, PA during housework, gardening, transportation, or work, suggesting that only PA of vigorous intensity should be included in the health recovery plan. In addition, the time spent sitting should also be reduced in the health recovery plan, as it has already been confirmed that sedentary behavior has a negative effect on health, regardless of whether the person has a good PA during the week [16]. 

Similarly, Peterson et al., found [19] that university students may be both highly active and highly sedentary. However, in their research they did not use IPAQ nor explored type or intensity of PA. Given the trend toward increased time in sedentary behaviors and since the relationship of many determinants with sedentary behavior remains inconsistent [20], there is still a need for more longitudinal research on determinants of sedentary behavior in young people. 

### 4.2. Association of PA and Sedentary Behavior (Time Spent Sitting) with BMI and Body Composition (Body Fat and Muscle Mass)

Although it would be expected that body fat percentage would be lowest in students in the Q4 [65], this was not the case in our study. However, the body fat percentage was the highest in Q1 in our study. In addition, there was a weak negative correlation found between PA of vigorous intensity and PA during recreation, leisure, and sports and body fat percentage. Moreover, the same PA variables were positively correlated with muscle mass. Similar results were reported by You et al. in students at a pre-university center in Malaysia [66]. Furthermore, no correlation between PA and BMI was found in our study, which is quite understandable since BMI is a poor indicator of body fat percentage in certain populations, such as college-age athletes and non-athletes [67] and individuals with a large body build [68]. 

There was also no correlation found between sitting duration and body fat or muscle mass. According to the systematic review of longitudinal studies by Huang et al., [20] there was insufficient evidence to find an association between sedentary behavior and adiposity indicators. Thus, additional longitudinal studies of high methodologic quality to clarify the relationships between sedentary behavior body composition among young adults are required.

### 4.3. Association of PA and Sedentary Behavior (Time Spent Sitting) with QoL and Sleep

In our study, a negative correlation was found between sitting duration and both the physical and psychosocial health categories of QoL. In contrast, PA correlated only with QoL related to physical health, but not with QoL related to psychosocial health. In addition, only PA of vigorous intensity and PA during recreation, leisure, and sports significantly correlated with QoL related to physical health. In a number of studies with college students or young adults, the results also showed a positive association between QoL and PA [13,14,15]. In our study, we did not analyze mental health specifically, but used only the psychosocial category of the PedQL^TM^ questionnaire. However, we found that psychosocial health, i.e., scores of QoL related to psychosocial health, was better in students in Q3 and Q4 quartiles with the highest PA than in the quartiles with the lowest PA (Q1 and Q2), but no significant differences in QoL related to psychosocial health were found between the PA quartiles. In a study by Jiang et al. [16], the results showed an association between sedentary behavior and mental well-being in college students in China, and this association may be partly due to impaired SQ. Sedentary behavior was positively associated with anxiety, depression, and suicidality in a dose-dependent manner, independent of PA. In a study by Ge et al. [17] sitting time was not associated with QoL in college students and in a study by Zhou et al. [18], this association was not entirely clear. There are many explanations for the relationship between PA and mental health, but regarding the relationship between sedentary lifestyle and mental well-being, the underlying mechanisms are still unclear [4,42,69,70,71,72,73,74].

A few studies have examined the association between sitting duration and SQ in college students [74,75,76], but the association remains insufficient, or sedentary behavior weakly negatively correlated with SQ. In our study, we found no correlation between time spent sitting and SQ. However, our results showed that SQ improved when PA was increased during recreation, leisure, and sports. Although researchers in the study by Santos et al. [77] did not use an IPAQ-LF self-report questionnaire that measures hours per week, they reached similar results, in which SQ was in association with PA during free time. However, there were no data on the type or intensity of PA, only on the frequency of PA during leisure time. Furthermore, in a study by Štefan et al., the results showed that poor SQ is associated with insufficient physical activity in young adults [30]. However, this study did not analyze any association between the type or intensity of PA and SQ, and did not use variables in MET-min/wk, but only the minimum recommended PA in minutes per week according to the recommendations of WHO [3,5,64]. 

### 4.4. Association of PA and Sedentary Behavior (Time Spent Sitting) with Posture (Trunk Muscle Endurance, Trunk Extensor/Flexor Endurance Ratio and Spinal Curvatures)

Prolonged sitting, especially among students, has been shown to be associated with worsening health status, regardless of PA [78]. Moreover, prolonged sitting is associated with poorer posture [62] and, subsequently, with imbalanced trunk muscles and abnormal spinal curvatures. It is known that an increase in thoracic kyphosis causes lumbar hyperlordosis to maintain sagittal balance [79]. This happens because fatigue-induced reduction in active muscle stiffness increases antagonistic co-contraction to maintain stability, resulting in increased spinal compression with fatigue [80]. Moreover, increasing spinal curvatures also shorten spinal musculature, which, in turn, negatively affects muscles, attempting to resist increased spinal load [22]. In our study, regardless of PA, participants were found to have borderline hyperkyphosis in all 3 quartiles with high PA (Q2, Q3, and Q4) with a mean value of 46.5 ± 8.7. No hypokyphotic students and very few hypolordotic students were found. Most students were hyperkyphotic (N = 49; 58.9%), but no significant differences in hyperkyphosis were found between the PA quartiles. In addition, most students with neutral lordosis were found in Q4 (N = 21; 25%). In some studies [62,79], hyperlordosis was found to be a more characteristic of young female students, whereas hyperkyphosis was more commonly in male students. In our study of 19 hyperlordotic students, 17 of them were female and 22 of 49 hyperkyphotic students were male, which is consistent with these studies. It is already known that an imbalance of the trunk extensors and trunk flexors can change the curvature of the lumbar spine and cause health problems [22,53,80]. In our study, lumbar lordosis angle correlated with trunk extensor endurance test scores, which may indicate imbalanced trunk muscle endurance in college students [23,24,25,26]. It seems the greater the lumbar lordosis, the greater the trunk extensor endurance test scores.

In a study by Kim et al. [53], the strength of lumbar flexor and extensor muscles was not significantly related to lordotic angle in both men and women. Only the ratio of extension to flexion showed a significant but very moderate relationship with lordotic angle. In our study, we had a similar result. We did not find any correlations between kyphotic and lordotic angles and accompanying flexor and extensor endurance test values. However, we used isometric trunk endurance tests to present their ratio, i.e., trunk extensor/flexor test ratio, and thus analyzed the correlation between the ratio and lordosis angle. Again, we found that the trunk extensor/flexor test ratio was highest among students in Q1 and significantly different from those in quartiles Q2 and Q3 quartiles, indicating that the students with the lowest PA had the greatest trunk muscle imbalance. Moreover, these were the students spending most their time spent sitting. Like the study by Taspinar et al. [81], we found that spinal curvatures of college students were not correlated with BMI but were positively correlated with body fat and negatively correlated with muscle mass. In our study, it was found that only lordosis was positively correlated with body fat and negatively correlated with muscle mass. Therefore, it can be concluded that as body fat percentage increases and muscle mass decreases, the lordosis angle increases.

In a study by Mirbagheri et al. [82], the correlation between lumbar lordosis and low back pain was statistically significant (*p* = 0.006), suggesting the need for a different questionnaire or visual analogue scale for pain for this type of study. In a study by Hamaoka et al. [83], the results showed a significant positive correlation between chronic low back pain and insomnia scores in college students. More specifically, a sedentary posture deforms the lumbar spine and increases internal pressure on the lumbar region, which is exacerbated by prolonged sitting [84]. Accordingly, college students are exposed to constant mechanical stress on their lower back and are therefore at higher risk of suffering from chronic lower back pain. However, the association between sedentary behavior and lower back pain, as well as lower back pain and insomnia should be investigated in future research.

### 4.5. Association of PA and Sedentary Behavior (Time Spent Sitting) with Cardiorespiratory Performance (Physical Fitness Index and MVV) 

For cardiorespiratory performance, we measured cardiovascular fitness (physical fitness index) and MVV. The results showed that physical fitness index scores were higher the more active the students were, but in general, most students had a poor cardiovascular fitness (N = 34; 41.7%). According to a systematic review by Kljajević et al. [85], it could be concluded that university students show a satisfactory level of physical fitness. However, the results vary because of different factors involved, mostly related to cultural differences and varying educational systems in different countries.

In addition, MVV correlated with PA during recreation, leisure, and sports, which is consistent with other studies. However, to our knowledge, no literature compares type of PA with MVV. Trained individuals tend to have higher MVV, operating at higher proportions of their MVV [86]. Untrained and sedentary individuals tend to have lower MVV as well as lower respiratory capacity due to lower oxygen uptake during exercise and lower capacity for physical work [87]. Our data showed no correlation between time spent sitting and MVV. However, we found a negative correlation between MVV and lumbar lordosis angle, suggesting that changes in lumbar spine and possibly lower back musculature are related to respiratory muscle performance, as we have discussed previously.

## 5. Strengths and Limitations

The study has strengths and limitations.

The strengths are as follows: (1) The main strength is the complexity of the assessments that is difficult to obtain. Our study is the first to examine and provide a more comprehensive physical health status of young adults. (2) We used the most up-to-date recommended methods for this type of study. (3) We assessed PA using the long form of the IPAQ test, dividing the data into quartiles of PA levels in MET -min/wk and comparing the independent variables (gender, BMI, body composition, PA level, PA in min/wk, time spent sitting, SQ, cardiovascular fitness, MVV, QoL, trunk muscle endurance, and spinal curvatures) by group. Additionally, we correlated different IPAQ-LF domains (vigorous intensity PA, PA during recreation, leisure, and sports, and time spent sitting) as dependent variables with all other independent variables.

The limitations are as follows: (1) The limitations of this study include its cross-sectional design with a convenience sample of young adult volunteers. Therefore, statements regarding cause and effect cannot be made. (2) One of the main limitations is the nature of the study, i.e., multiple measurements, which requires participants to make extra effort to complete all tests. It was rather challenging to get participants to complete all tests. Thus, although the initial number of participants was 126, in the end there were 82 students who completed all the required tests. This could be the reason why the current study was not sufficiently powered to establish statistical significance of the association of PA and sedentary behavior with, for example, spinal curvatures, QoL related to psychosocial health, and cardiovascular fitness. Furthermore, the correlation coefficients were significant but weak. The number of participants would need to be higher, which we intend to do in the future. (3) This study did not measure associations between independent variables, e.g., association of SQ with QoL, association of body composition (percentage of body muscle mass and body fat) and BMI with trunk muscle endurance, cardiovascular fitness, MVV, QoL or SQ, association of muscle endurance, cardiovascular fitness, MVV with SQ and QoL. (4) The students were predominantly from the Faculty of Health Studies, Medicine, and Maritime Studies, who may be more physically active than students from other faculties such as the Faculty of Law and the Faculty of Philosophy. In addition, we did not include young adults who do not study. (5) Female subjects outnumbered male subjects (F:M = 48:34). In addition, we did not fully investigate whether the results differed between the sexes. (6) We also did not analyze dietary habits. (7) We did not include the VO_2_-max test and other types of tests to assess physical condition, or questionnaires to assess mental health in detail in college students. The study also lacked an objective measurement of PA. Additional prospective studies using the IPAQ in combination with objective measures of PA are needed to confirm the findings of this study. (8) The device SM has its limitations, as it only measures the thoracic and lumbar spine, but not the cervical part of the spine. Future research should include a larger sample to generalize the results and include different types of achievement tests, more specific quality of life questionnaires and more association analyses between variables. 

## 6. Conclusions

The results of our study showed positive associations between PA and QoL related to physical health, MVV, and muscle mass, and furthermore, a negative association between time spent sitting and overall QoL, body fat, and SQ in college students. Although this was not the primary goal of this study, the results also showed a positive relationship between lumbar lordosis and body fat, the trunk extensor test, trunk extensor/flexor test ratio, muscle mass, and MVV. Although it had a smaller sample size, in our study we found sufficient evidence to correlate PA/inactivity and time spent sitting with trunk muscle endurance, QoL, SQ, respiratory function, and body composition, as well as to correlate changes in spinal curvatures with trunk extensor endurance, trunk extensor/flexor test ratio, body composition, and MVV. Moreover, our results suggest that students with low PA levels and more time spent sitting have imbalanced trunk muscles, worse respiratory function, as well as QoL and sleep. These relationships between the above variables in college students could reflect their student lifestyle. These results highlight the importance of young adult participation in PA, as the transition from adolescence to adulthood may be the last stage of life in which lifestyle habits can be monitored and influenced, as they have a major impact on future life. Awareness of the importance of PA should be raised among students or other young adults so that it not only leads to prevention of chronic cardiovascular, metabolic, and musculoskeletal diseases and improved QoL in adulthood, but also leads to positive changes in society and, thus, additional return on investment. Accordingly, it is critical to incorporate active lifestyle activities into school curricula to counteract sedentary-related problems and issues associated with physical inactivity. Replacing sedentary time with PA of any intensity in everyday life, including light physical activities, provides health benefits; that is, by choosing to go by foot or bicycle to school or work instead of by car or bus, or choosing to go by stairs instead of lift or escalator, etc. 

## Figures and Tables

**Figure 1 ijerph-20-06938-f001:**
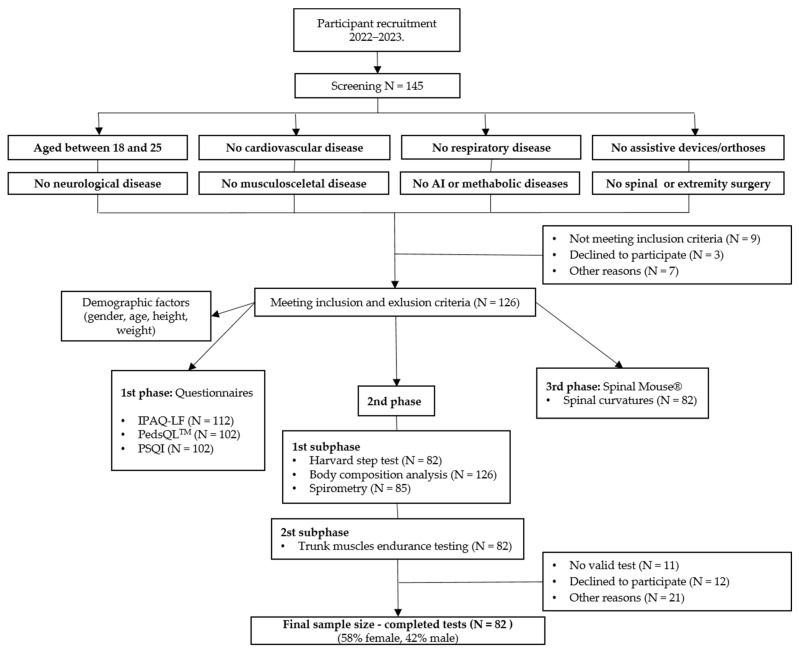
Flowchart of number of participants at different stages in the study. AI, autoimmune; N, number; IPAQ-LF, the International Physical Activity Questionnaire-Long-Form; PedsQL™, Pediatric Quality of Life Inventory™; PSQI, Pittsburgh Sleep Quality Index.

**Table 1 ijerph-20-06938-t001:** Descriptive data from the different domains of the International Physical Activity Questionnaire-Long Form (IPAQ-LF) (N = 82).

IPAQ-LF Domains	Mean ± SE
**Type of PA (MET-min/wk)**	
PA at work (MET-min/wk) PA during transport (MET-min/wk) PA at home or in garden (MET-min/wk) Recreation, sport, and leisure-time PA (MET-min/wk)	1176.7 ± 259.4 926.3 ± 104.7 1002.2 ± 155.3 2469.6 ± 297.7
**Intensity of PA (MET-min/wk)**	
Walking (MET-min/wk) Moderate PA (MET-min/wk) Vigorous PA (MET-min/wk)	2184.8 ± 222.6 1523.3 ± 198.0 1865.4 ± 307.3
**Total PA (MET-min/wk)**	5573.9 ± 494.4
**Time spent sitting (min/wk)**	2489.7 ± 105.1

SE, standard error; PA, Physical Activity; MET, Metabolic Equivalent of Task; wk, week.

**Table 2 ijerph-20-06938-t002:** Percentage of participants in PA quartiles by gender, BMI, body composition, PA, sleep quality, cardiovascular fitness, and respiratory function.

Variable	Quartiles of Self-Reported PA (MET-min/wk)					
	Q1 (672–2087) N (%)	Q2 (2088–4008) N (%)	Q3 (4009–8573) N (%)	Q4 (8574–22,062) N (%)	Total N (%)	*p* Value
**Gender**						
Female Male Total	16 (19.5%) 3 (3.7%) 19 (23.2%)	11 (13.4%) 9 (11.0%) 20 (24.4%)	10 (12.2%) 11 (13.4%) 21 (25.6%)	11 (13.4%) 11 (13.4%) 22 (26.8%)	48 (58.5%) 34 (41.5%) 82 (100.0%)	^a^ (df = 3) 0.703
**^b^ BMI (kg/m^2^)**						
Normal Overweight Obese	15 (18.3%) 4 (4.9%) 0 (0.0%)	15 (18.3%) 3 (3.7%) 2 (2.5%)	13 (15.8%) 7 (8.5%) 1 (1.2%)	14 (17.1%) 7 (8.5%) 1 (1.2%)	57 (69.5%) 21 (25.6%) 4 (4.9%)	^a^ (df = 3) 0.549
**Body composition (Tanita RD-545)**						
^c^ Visceral fat level						
Healthy level rating	19 (23.2%)	20 (24.4%)	21 (25.6%)	22 (26.8%)	82 (100.0%)	
Body fat (%)						
Underfat Healthy Overweight Obese	0 (0.0%) 14 (17.1%) 5 (6.1%) 0 (0.0%)	1 (1.2%) 15 (18.3%) 2 (2.4%) 2 (2.4%)	0 (0.0%) 14 (17.1%) 5 (6.1%) 2 (2.4%)	1 (1.2%) 13 (15.9%) 5 (6.1%) 3 (3.7%)	2 (2.4%) 56 (68.4%) 17 (20.7%) 7 (8.5%)	^a^ (df = 9) 0.708
Muscle mass (%)						
Very low Low Good Increased	0 (0.0%) 3 (3.7%) 16 (19.5%) 0 (0.0%)	2 (2.4%) 1 (1.2%) 16 (19.5%) 1 (1.2%)	2 (2.4%) 3 (3.7%) 16 (19.5%) 0 (0.0%)	3 (3.7%) 2 (2.4%) 15 (18.3%) 2 (2.4%)	7 (8.5%) 9 (11.0%) 63 (76.8%) 3 (3.7%)	^a^ (df = 9) 0.594
**Physical activity (IPAQ-LF)**						
Minimum required PA per week						
Sufficient Insufficient	15 (18.3%) 4 (4.9%)	1 (1.2%) 19 (23.2%)	0 (0.0%) 21 (25.6%)	1 (1.2%) 21 (25.6%)	17 (20.7%) 65 (79.3%)	^a^ (df = 3) 0.000 *
^d^ PA level						
Minimally active HEPA active	15 (18.3%) 4 (4.9%)	10 (12.2%) 10 (12.2%)	10 (12.2%) 11 (13.4%)	6 (7.3%) 16 (19.5%)	41 (50.0%) 41 (50.0%)	^a^ (df = 3) 0.012 *
**Sleep quality (PSQI)**						
Good sleep quality Poor sleep quality	3 (3.7%) 16 (19.5%)	12 (14.6%) 8 (9.7%)	7 (8.5%) 14 (17.0%)	9 (11.0%) 13 (15.8%)	31 (37.8%) 51 (62.2%)	^a^ (df = 3) 0.038 *
**Cardiovascular fitness (HST)**						
Poor Below average Average Good Excellent	10 (11.7%) 0 (0.0%) 5 (6.7%) 1 (1.7%) 1 (1.7%)	8 (10.0%) 3 (3.3%) 5 (6.7%) 4 (5.0%) 0 (0.0%)	8 (10.0%) 1 (1.7%) 11 (13.3%) 0 (0.0%) 1 (1.7%)	8 (10.0%) 3 (3.3%) 3 (3.3%) 5 (6.7%) 3 (3.3%)	34 (41.7%) 7 (8.3%) 25 (30.0%) 11 (13.3%) 5 (6.7%)	^a^ (df = 12) 0.269
**MVV (L/min) (Spirometry)**						
Below normal Normal	0 (0.0%) 18 (21.7%)	1 (1.7%) 19 (23.1%)	2 (2.1%) 19 (23.1%)	1 (1.7%) 22 (26.7%)	4 (5.4%) 78 (94.6%)	^a^ (df = 3) 0.613

^a^ Chi-square test; ^b^ there were no underweight and no extremely obese students registered; ^c^ there were no students with excess levels of visceral fat; ^d^ there were no inactive students registered; BMI, Body Mass Index; HEPA, health-enhancing PA; MET, Metabolic Equivalent of Task; wk, week; Q, Quartiles; N, number; df, degree of freedom for error; BMI, Body Mass Index; PA, Physical Activity; IPAQ-LF, the International Physical Activity Questionnaire-Long Form; PedsQL^TM^, Pediatric Quality of Life Inventory^TM^; HST, Harvard Step Test; MVV, Maximal Voluntary Ventilation. * statistical significance.

**Table 3 ijerph-20-06938-t003:** Percentage of participants in PA quartiles by values of thoracic kyphosis and lumbar lordosis in straight posture.

Variable	Quartiles of Self-Reported PA (MET-min/wk)					
	Q1 (672–2087) N (%)	Q2 (2088–4008) N (%)	Q3 (4009–8573) N (%)	Q4 (8574–22,062) N (%)	Total N (%)	*p* Value
**Spinal curvatures (Spinal Mouse** **^®^)**						
**^a^ Thoracic kyphosis angle–straight posture (°)**						
Neutral kyphosis Hyperkyphosis	7 (8.9%) 12 (14.3%)	10 (12.6%) 9 (10.7%)	7 (8.9%) 13 (16.1%)	9 (10.7%) 15 (17.8%)	33 (41.1%) 49 (58.9%)	^b^ (df = 3) 0.697
**Lumbar lordosis angle–straight posture (°)**						
Hypolordosis Neutral lordosis Hyperlordosis	0 (0.0%) 13 (16.0%) 6 (7.1%)	0 (0.0%) 14 (17.9%) 4 (5.4%)	3 (3.6%) 12 (14.3%) 6 (7.1%)	0 (0.0%) 21 (25.0%) 3 (3.6%)	3 (3.6%) 60 (73.2%) 19 (23.2%)	^b^ (df = 6) 0.124

^a^ there were no hypokyphotic students registered; ^b^ Chi-square test. MET, Metabolic Equivalent of Task; wk, week; Q, Quartiles; N, number; df, degree of freedom for error.

**Table 4 ijerph-20-06938-t004:** Characteristic of subjects and comparison of variables between quartiles of self-reported PA.

Variable	Quartiles of Self-Reported PA (MET-min/wk)					
	Q1 (672–2087) Mean ± SD or Median (Range)	Q2 (2088–4008) Mean ± SD or Median (Range)	Q3 (4009–8573) Mean ± SD or Median (Range)	Q4 (8574–22,062) Mean ± SD or Median (Range)	Total Mean ± SD or Median (Range)	*p* Value
**BMI (kg/m^2^)**	22.6 ± 2.5	23.5 ± 3.2	23.9 ± 2.6	24.9 ± 3.2	23.8 ± 2.9	^a^ Q1:Q2 = 0.781 Q1:Q3 = 0.560 Q1:Q4 = 0.089 Q2:Q3 = 0.982 Q2:Q4 = 0.457 Q3:Q4 = 0.672
**Body composition** **(Tanita RD-545)**						
Visceral fat level	1.5 (1–4.5)	2 (1–10)	2.5 (1–4.5)	3 (1–11)	2 (1–11)	^b^ 0.311
Body fat (%)	27.2 ± 5.8	24.6 ± 7.7	25 ± 8.3	24.6 ± 9.2	25.3 ± 7.8	^a^ Q1:Q2 = 0.751 Q1:Q3 = 0.835 Q1:Q4 = 0.750 Q2:Q3 = 0.998 Q2:Q4 = 1.000 Q3:Q4 = 0.998
Muscle mass (%)	68.9 ± 5.4	71.6 ± 7.3	71.2 ± 7.8	71.6 ± 8.8	70.9 ± 7.4	^a^ Q1:Q2 = 0.710 Q1:Q3 = 0.793 Q1:Q4 = 0.710 Q2:Q3 = 0.998 Q2:Q4 = 1.000 Q3:Q4 = 0.998
Physique rating	5 (1–5)	5 (2–8)	5 (2–6)	5 (2–8)	5 (1–8)	^a^ Q1:Q2 = 0.671 Q1:Q3 = 0.978 Q1:Q4 = 0.909 Q2:Q3 = 0.876 Q2:Q4 = 0.963 Q3:Q4 = 0.992
**Physical activity (IPAQ-LF)**						
PA (MET-min/wk)	1475 ± 421	3010 ± 567	5559 ± 1059	11,459 ± 4252	5574 ± 4478	^a^ Q1:Q2 = 0.000 * Q1:Q3 = 0.000 * Q1:Q4 = 0.000 * Q2:Q3 = 0.004 * Q2:Q4 = 0.000 * Q3:Q4 = 0.000 *
Time spent sitting (min/wk)	2856 ± 1060	2629 ± 658	2477 ± 1001	2059 ± 924	2490 ± 952	^a^ Q1:Q4 = 0.046 *
PA level	2 (2–3)	2 (2–3)	3 (2–3)	3 (2–3)	2 (2–3)	^b^ 0.027 *
**Quality of life (PedsQL^TM^)**						
Physical health	79.3 ± 12.2	82.7 ± 11.6	89.7 ± 8.7	87.1 ± 11.1	84.9 ± 11.5	^a^ Q1:Q2 = 0.776 Q1:Q3 = 0.021 * Q1:Q4 = 0.133 Q2:Q3 = 0.180 Q2:Q4 = 0.580 Q3:Q4 = 0.860
Psychosocial health	78.9 ± 11.4	81.9 ± 11.5	81.5 ± 11.2	82.3 ± 9.8	81.2 ± 10.8	^a^ Q1:Q2 = 0.825 Q1:Q3 = 0.876 Q1:Q4 = 0.773 Q2:Q3 = 0.999 Q2:Q4 = 0.999 Q3:Q4 = 0.996
Total score	79.0 ± 10.1	82.2 ± 10.3	84.4 ± 9.5	83.9 ± 8.8	82.5 ± 9.7	^a^ Q1:Q2 = 0.774 Q1:Q3 = 0.324 Q1:Q4 = 0.400 Q2:Q3 = 0.888 Q2:Q4 = 0.938 Q3:Q4 = 0.998
**Sleep quality (PSQI)**	6 (2–11)	4 (2–17)	5 (2–8)	5 (2–8)	5 (2–17)	^b^ 0.055
**Trunk muscle endurance**						
Trunk flexor test (sec)	166.8 ± 32.4	174.7 ± 20.4	178.2 ± 7.0	180.0 ± 0.0	172.1 ± 21.8	^a^ Q1:Q2 = 0.149 Q1:Q3 = 0.253 Q1:Q4 = 0.149 Q2:Q3 = 0.990 Q2:Q4 = 1.000 Q3:Q4 = 0.988
Trunk extensor test (sec)	176.2 ± 13.8	136.3 ± 40.1	150.5 ± 37.8	152.1 ± 45.1	152.9 ± 38.6	^a^ Q1:Q2 = 0.039 * Q1:Q3 = 0.300 Q1:Q4 = 0.355 Q2:Q3 = 0.722 Q2:Q4 = 0.652 Q3:Q4 = 0.999
Trunk extensor/flexor test ratio	1.1 ± 0.2	0.7 ± 0.2	0.8 ± 0.2	0.9 ± 0.3	0.9 ± 0.2	^a^ Q1:Q2 = 0.014 * Q1:Q3 = 0.031 * Q1:Q4 = 0.066 Q2:Q3 = 0.914 Q2:Q4 = 0.427 Q3:Q4 = 0.802
**Spinal curvatures (Spinal Mouse** **^®^)**						
Thoracic kyphosis angle–straight posture (°)	44.7 ± 10.3	47.0 ± 9.1	47.2 ± 6.1	46.9 ± 9.0	46.5 ± 8.7	^a^ Q1:Q2 = 0.919 Q1:Q3 = 0.896 Q1:Q4 = 0.925 Q2:Q3 = 0.999 Q2:Q4 = 0.999 Q3:Q4 = 0.999
Lumbar lordosis angle–straight posture (°)	−34.9 ± 7.5	−32.1 ± 8.6	−32.0 ± 11.8	−32.3 ± 9.2	−32.8 ± 9.2	^a^ Q1:Q2 = 0.868 Q1:Q3 = 0.858 Q1:Q4 = 0.894 Q2:Q3 = 0.999 Q2:Q4 = 0.999 Q3:Q4 = 0.999
**Cardiovascular fitness (HST)**	54 ± 25	59 ± 19	60 ± 18	65 ± 22	59 ± 21	^a^ Q1:Q2 = 0.910 Q1:Q3 = 0.856 Q1:Q4 = 0.499 Q2:Q3 = 0.999 Q2:Q4 = 0.854 Q3:Q4 = 0.903
**MVV (L/min) (Spirometry)**	133.0 ± 35	153 ± 30	150 ± 31	151 ± 27	147 ± 31	^a^ Q1:Q2 = 0.325 Q1:Q3 = 0.479 Q1:Q4 = 0.447 Q2:Q3 = 0.991 Q2:Q4 = 0.995 Q3:Q4 = 0.999

^a^ one-way ANOVA post-hoc Tukey test; ^b^ Kruskal–Wallis test; MET, Metabolic Equivalent of Task; wk, week; Q, Quartiles; SD, standard deviation; BMI, Body Mass Index; PA, Physical Activity; IPAQ-LF, the International Physical Activity Questionnaire-Long Form; PedsQL^TM^, Pediatric Quality of Life Inventory^TM^, PSQI, Pittsburgh Sleep Quality Index; HST, Harvard Step Test; MVV, Maximal Voluntary Ventilation. * statistical significance.

**Table 5 ijerph-20-06938-t005:** Pearson correlation analyses for PA and time spent sitting vs. BMI, body composition, quality of life and sleep, trunk muscles endurance, cardiovascular fitness, respiratory function, and spinal curvatures.

N = 82	IPAQ-LF Measures			
Variable	Vigorous PA (MET-min/wk) r/p	PA during Recreation, Sport and Leisure-Time (MET-min/wk) r/p	Total PA (MET-min/wk)r/p	Time Spent Sitting (min/wk) r/p
**BMI (kg/m^2^)**	0.095/0.393	0.182/0.101	0.181/0.102	−0.149/0181
**Body composition** **(Tanita RD-545)**				
Visceral fat level	0.199/0.859	0.121/0.276	0.077/0.490	−0.144/0.195
Body fat (%)	−0.254/0.021 *	−0.278/0.011 *	−0.158/0.156	0.041/0.716
Muscle mass (%)	0.260/0.018 *	0.287/0.009 *	0.165/0.137	−0.078/0.486
Physique rating	0.154/0.166	0.111/0.318	0.101/0.364	−0.010/0.929
**Quality of life (PedsQL^TM^)**				
Physical health	0.374/0.001 *	0.416/0.000 *	0.301/0.006 *	−0.246/0.026 *
Psychosocial health	0.061/0.586	0.132/0.236	0.102/0.361	−0.247/0.025 *
Total score	0.197/0.075	0.266/0.015 *	0.197/0.075	−0.280/0.011 *
**Sleep quality (PSQI)**	−0.125/0.263	−0.243/0.028 *	−0.135/0.226	0.142/0.202
**Trunk muscle endurance**				
Trunk flexor test (sec)	0.041/0.710	−0.033/0.764	−0.027/0.805	−0.081/0.464
Trunk extensor test (sec)	0.070/0.530	−0.013/0.907	−0.006/0.956	−0.058/0.604
Trunk extensor/flexor test ratio	−0.067/0.546	0.002/0.980	0.0107/0.924	0.056/0.616
**Spinal curvatures (Spinal Mouse** **^®^)**				
Thoracic kyphosis angle-straight posture (°)	0.062/0.648	0.095/0.484	0.081/0.557	0.012/0.925
Lumbar lordosis angle-straight posture (°)	−0.006/0.962	−0.078/0.566	−0.040/0.765	−0.005/0.967
**Cardiovascular fitness (HST)**	0.199/0.140	0.221/0.100	0.214/0.100	−0.093/0.493
**MVV (L/min) (Spirometry)**	0.175/0.196	0.341/0.010 *	0.169/0.195	−0.053/0.697

* statistical significance; PA, Physical Activity; BMI, Body Mass Index; N, number; IPAQ-LF, the International Physical Activity Questionnaire-Long Form; MET, Metabolic Equivalent of Task; wk, week; r, Pearson correlation coefficient; *p*, level of statistical significance; PSQI, Pittsburgh Sleep Quality Index; PedsQL^TM^, Pediatric Quality of Life Inventory^TM^; HST, Harvard Step Test; MVV, Maximal Voluntary Ventilation.

## Data Availability

Data supporting this article are available from the corresponding author on reasonable request.

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
