# Peer review of "Relationship between Physical Activity and Sedentary Behavior, Spinal Curvatures, Endurance and Balance of the Trunk Muscles-Extended Physical Health Analysis in Young Adults"

_ijerph, 2023, doi:10.3390/ijerph20206938_

Round 1

Reviewer 1 Report

·         What is the novelty of this study? The association of physical activity and sedentary behaviour with the parameters has been studied several times. The authors themselves quoted these types of studies in the discussion.  

·         Line 74- 76 – This statement is not proper

·         In the introduction section, the authors quoted only studies from Croatia. Is it a regional study?

·         The introduction does not have enough literature review, and the study's rationale is unclear.

·         Why the authors did not calculate the sample size

·         The arrangement of the tables is not in proper order. Please follow the journal guidelines about the position of the table inside the text.

·         Line 221-238- Is it a footnote of the table?

·         The result and discussion are properly written. 

Author Response

Reviewer 1:

Comments and Suggestions for Authors

Dear Reviewer, thank you for taking the time to provide us with useful comments and suggestions to improve our work. We have taken your advice and have revised everything in detail according to your advice. We sincerely hope that you will be satisfied with these changes. Sincerely, the authors. All our responses can be found below:

Reviewer: What is the novelty of this study? The association of physical activity and sedentary behavior with the parameters has been studied several times. The authors themselves quoted these types of studies in the discussion.  

Response: we thank the reviewer for the constructive comments. We have explained our aim in more detail in the abstract and introduction. There is no study that examines physical activity and sedentary behavior with endurance and balance of the trunk muscles  and spinal curvatures (kyphosis and lordosis), especially in young adults aged between 18-25 years. Furthermore, there is no study that examines a broader physical health status related to physical activity and sedentary behavior (highlighted).

Reviewer: Line 74- 76 – This statement is not proper

Response: we thank the reviewer. We have changed the introduction part and deleted this sentence accordingly (highlighted).

Reviewer: In the introduction section, the authors quoted only studies from Croatia. Is it a regional study? The introduction does not have enough literature review, and the study's rationale is unclear.

Response: Yes, a little bit. We agree with you completely. We have now changed the introduction, linked all the variables we measured and cited more international work. We have added a scheme as supplementary Figure S1 showing the interplay of possible consequences of low physical activity and sedentary behavior with the associated measurement methods to facilitate understanding of the variables included in the study (highlighted).

Reviewer: Why the authors did not calculate the sample size

Response: we thank the reviewer for the comments. Yes, we have done the calculation. However, we did not explain it. Now we have included it in the Methods section (lines: 133-155, highlighted), added a flow chart (Figure 1) and explained a limitation of the study in the Limitation section (lines: 733-759, highlighted).

Reviewer: The arrangement of the tables is not in proper order. Please follow the journal guidelines.

Response: We thank the reviewer for the useful comments. Now we have revised the order of the tables according to the guidelines of the journal. We have rearranged the text on the results so that it follows each table (lines: 385-559, highlighted).

Reviewer: Line 221-238- Is it a footnote of the table?

Response: We removed this footnote to the Method section and explained all variables (lines: 169-383, highlighted).

Reviewer: The result and discussion are properly written. 

Response: we have arranged the results so that they follow each table for clarity. We have also separated the discussion section with headings to make it clearer (lines: 384-720, highlighted).

Reviewer 2 Report

Association of physical activity and sedentary behavior with physiological spinal curvatures, trunk muscle endurance, cardiorespiratory performance, body composition and quality   life and sleep in young adults – a cross-sectional analysis

Marijančić, etal, 2023

I want to thank the authors and the Editors for allowing me the opportunity to evaluate their work, which was submitted to a respectable Int. J. Environ. Res. And Public Health. The authors conducted a survey study to assess the extent to which PA levels and sedentary lifestyle are associated with changes in the musculoskeletal system, cardiorespiratory performance, QoL and sleep in young adults. The topic is interesting for the Journal readers. Although the authors exerted much effort on this work, the manuscript in its current situation is very difficult to follow, and the relationship between the different variables needs to be better articulated and provide the justification for why the authors focused on all these domains. My concerns are the following

Title:

 Very long title should be improved to be short, easy to understand, and convey the research's important aspects.

1.     Abstract:

 - Your Abstract should answer these questions: What was done? Why did you do it? What did you find? Why are these findings useful and important? Suggestions for improvement,

- Please do not provide unnecessary “details in the abstract section 

- Please reduce the results in the abstract and keep the most important.

- No valid conclusion or public health implication is provided in the abstract. 

2.     Introduction:

        It is conceptually not clear how the different elements (physical activity and sedentary behavior, physiological spinal curvatures, trunk muscle endurance, cardiorespiratory performance, body composition and quality of life) that you assessed are related to each other in the context of your study and the rationale you have used to link these (supported by international literature).

        Each study is developed based on research questions or hypotheses. The authors should clearly indicate their research questions in the introduction after highlighting their study contributions

Methods:

The methodology section needs to be improved as it is in its current situation is misleading and specifically the following  

  • All sections of methods should be rewritten following STROBE guidelines, eg many sections of the methods provided a mixture of information (methods and Results at the same time) that confuse readers.
  • The study setting and population need to be better defined—no specific date for the survey is provided.
  • The sample size and design need clarification; why only 82 students, and how do you select them? What margin of error? All inferential statistics used should be understood in the context of the sampling parameters. This further will help in understanding the accuracy of the study outcomes.
  • The authors used many international tools and are well known, so there is no need for a detailed description of the questionnaires. The whole section of the questionnaires needs to be improved and summarized.  
  • A section about study variables and their definitions is essential.
  • There is no need to present any results in the methods section; the mixing between both affected the manuscript negatively. For example, in line 209, "Full details of body composition are explained in the legend of Table 1", any information related to the methods should be presented in the methods section.
  • Another mixing is found in the statistical analysis section. Information presented in Table 2 should be included in the Results section with the background characteristics. 

Results section:

Very difficult to follow, many results are presented without a clear goal. Even the organization is lacking; for example, the authors describe Table 5, and after that gain, they returned to Table 1 and Table 4. Results should be described consistently and reflect the study objectives and Hypothesis. Some categories in the Tables are 0(0%) zeros; there is no need to present all of them; instead, you can make a note in the Tables legend. Also, how is the Chi-squared test calculated for Rows with zero-sum? The symbol (– n) in Table 3 is not clear.

Discussion;

Unable to Judge until a complete revision of the methods and results section is conducted 

Limitations:

Not enough and needs to be comprehensive; what about sampling selection issues? and the study design? (Observational cross-sectional study design) etc. 

Conclusion:

It is well prepared. But for improvement, the conclusion section should also clearly restate the study's significance and contributions. Highlighting the importance of young adult participation in PA is valid, but specific recommendations could be included.

Author Response

Reviewer 2:

Comments and Suggestions for Authors

Association of physical activity and sedentary behavior with physiological spinal curvatures, trunk muscle endurance, cardiorespiratory performance, body composition and quality   life and sleep in young adults – a cross-sectional analysis

Marijančić, etal, 2023

 I want to thank the authors and the Editors for allowing me the opportunity to evaluate their work, which was submitted to a respectable Int. J. Environ. Res. And Public Health. The authors conducted a survey study to assess the extent to which PA levels and sedentary lifestyle are associated with changes in the musculoskeletal system, cardiorespiratory performance, QoL and sleep in young adults. The topic is interesting for the Journal readers. Although the authors exerted much effort on this work, the manuscript in its current situation is very difficult to follow, and the relationship between the different variables needs to be better articulated and provide the justification for why the authors focused on all these domains. My concerns are the following

Dear Reviewer, thank you for taking the time to provide us with useful comments and suggestions to improve our work. We have taken your advice and have revised everything in detail according to your advice. We sincerely hope that you will be satisfied with these changes. All our responses can be found below:

Title:

Reviewer: Very long title should be improved to be short, easy to understand, and convey the research's important aspects.

Response: we thank the reviewer for the useful comment. We changed the title and hope this will be better (highlighted).

Abstract:

Reviewer: - Your Abstract should answer these questions: What was done? Why did you do it? What did you find? Why are these findings useful and important? Suggestions for improvement,

- Please do not provide unnecessary “details in the abstract section 

- Please reduce the results in the abstract and keep the most important.

- No valid conclusion or public health implication is provided in the abstract. 

Response: We thank the reviewer for the constructive comments. We have changed the abstract accordingly. We have removed all unnecessary details, reduced the results to include only what is most important and added a valid conclusion. We hope this is better now (highlighted).

Introduction:

Reviewer: It is conceptually not clear how the different elements (physical activity and sedentary behavior, physiological spinal curvatures, trunk muscle endurance, cardiorespiratory performance, body composition and quality of life) that you assessed are related to each other in the context of your study and the rationale you have used to link these (supported by international literature).

  • Each study is developed based on research questions or hypotheses. The authors should clearly indicate their research questions in the introduction after highlighting their study contributions

Response: We thank the reviewer for the useful comments. We have changed some parts of introduction; explained the elements that we assessed in the study, and explained their connections; included  scheme as supplementary Figure S1 showing the interaction of possible consequences of low physical activity and sedentary behavior with the associated measurement methods to facilitate understanding of the variables included in the study; included the main goals of the study and the hypothesis (highlighted).

Methods:

Reviewer: The methodology section needs to be improved as it is in its current situation is misleading and specifically the following  

All sections of methods should be rewritten following STROBE guidelines, eg many sections of the methods provided a mixture of information (methods and Results at the same time) that confuse readers.

Response: we thank the reviewer for useful comments. We have made changes to the methods section, removed all unnecessary parts from the results, made corrections to the questionnaire section, added additional explanations of the variables we analyzed in the study (they have been moved from Tables 1, 3 and 4), explained the sampling of participants, and added a flowchart showing the different stages of the study, all according to the guidelines of STROBE (lines: 133-380, highlighted). We have also uploaded a STROBE checklist into the system.

Reviewer: The study setting and population need to be better defined—no specific date for the survey is provided.

Response: we thank the reviewer for the useful comment. We have added a period for recruitment and included a flow chart (Figure 1) for better understanding (lines: 133-155, highlighted).

Reviewer: The sample size and design need clarification; why only 82 students, and how do you select them? What margin of error? All inferential statistics used should be understood in the context of the sampling parameters. This further will help in understanding the accuracy of the study outcomes.

Response: we thank the reviewer for the useful comment. We have changed these parts in the methods (participants and study design) (lines: 133-155, highlighted), added an explanation of how sample size is calculated and explained in the manuscript limitations section (lines: 733-759, highlighted), added a flowchart for better understanding and uploaded the checklist STROBE to the system.

Reviewer: The authors used many international tools and are well known, so there is no need for a detailed description of the questionnaires. The whole section of the questionnaires needs to be improved and summarized.  

Response: we thank the reviewer for the useful comment. We changed this part with questionnaires (lines: 168-187, highlighted).

Reviewer: A section about study variables and their definitions is essential.

Response: we thank the reviewer for the useful comment. We have added a new heading: „Outcome measures“ and have moved all variable descriptions from the legends of Table 1. and Table 3. to the methods section and made them part of the individual measurements. We have also added further explanations for all variables (lines: 167-380, highlighted).

Reviewer: There is no need to present any results in the methods section; the mixing between both affected the manuscript negatively. For example, in line 209, "Full details of body composition are explained in the legend of Table 1", any information related to the methods should be presented in the methods section.

Response: we thank the reviewer for the useful comment. We deleted that part.

Reviewer: Another mixing is found in the statistical analysis section. Information presented in Table 2 should be included in the Results section with the background characteristics.

Response: we thank the reviewer for the useful comment. We moved that part to Results (highlighted).

Results section:

Reviewer:Very difficult to follow, many results are presented without a clear goal. Even the organization is lacking; for example, the authors describe Table 5, and after that gain, they returned to Table 1 and Table 4. Results should be described consistently and reflect the study objectives and Hypothesis. Some categories in the Tables are 0(0%) zeros; there is no need to present all of them; instead, you can make a note in the Tables legend. Also, how is the Chi-squared test calculated for Rows with zero-sum? The symbol (– n) in Table 3 is not clear.

Response: we thank the reviewer for the useful comment. We have rearranged the results to follow the table exactly according to Journal's directions.

Thank you for pointing out the chi-square test. We apologies for the misunderstanding. In the statistical analyses with chi-square test, we did not include the variables that do not exist at all (have zeros). We only added these zeros to the tables for informational purposes, but now we understand that this could be confusing for the reader. We have removed these zeros from the tables (1 and 3) and added an additional explanation in the footnote of these 2 tables and also in the statistical section.

We have removed “-“ and changed it into “N (%)” (highlighted).

Discussion;

Unable to Judge until a complete revision of the methods and results section is conducted 

Response: we added headings that separate the outcomes of the study so it can be easier to follow (highlighted).

Limitations:

Not enough and needs to be comprehensive; what about sampling selection issues? and the study design? (Observational cross-sectional study design) etc. 

Response: we thank the reviewer for the useful comments. We added more explanations to limitations. We have also added a strengths of the study (lines: 723-759, highlighted).

Conclusion:

It is well prepared. But for improvement, the conclusion section should also clearly restate the study's significance and contributions. Highlighting the importance of young adult participation in PA is valid, but specific recommendations could be included.

Response: we thank the reviewer for the useful comments. We added more comprehensive conclusion with study's significance and contributions, and more specific recommendations (highlighted).

Round 2

Reviewer 1 Report

According to the authors - “There is no study that examines physical activity and sedentary behavior with endurance and balance of the trunk muscles and spinal curvatures (kyphosis and lordosis), especially in young adults aged between 18-25 years. Furthermore, there is no study that examines a broader physical health status related to physical activity and sedentary behavior’

 if this is the reason for conducting the study, what is the reason for measuring musculoskeletal and cardiopulmonary health, quality of life, and sleep? These parameters have been studied several times.

The sample size calculation is not clear. What was the actual sample size required for this study? 77 or 123? 

Author Response

Reviewer 1:

Comments and Suggestions for Authors

Dear Reviewer,

we would like to thank you for your comments and the effort to improve our manuscript. Please find all our responses below:

Reviewer:

According to the authors - “There is no study that examines physical activity and sedentary behavior with endurance and balance of the trunk muscles and spinal curvatures (kyphosis and lordosis), especially in young adults aged between 18-25 years. Furthermore, there is no study that examines a broader physical health status related to physical activity and sedentary behavior’

 if this is the reason for conducting the study, what is the reason for measuring musculoskeletal and cardiopulmonary health, quality of life, and sleep? These parameters have been studied several times.

Response:

we thank the reviewer for the comment. We have explained in the abstract (lines: 26-29) and in the introduction part (lines: 119-122) that there is no study like ours that includes an extended analysis of physical health status in young adults aged 18-25 years, including measures of musculoskeletal health, which includes muscle mass, muscle endurance and spinal curvatures; cardiopulmonary health, which refers to physical fitness; and finally quality of life and sleep, which are final measures that direcly depend on musculoskeletal and cardiopulmonary health. While there are studies that have examined quality of life and sleep in relation to physical activity or sedentary behavior, there are no extended measures on one convenient sample group at the same time, in particular no analysis by PA quartiles (limitations: lines 726-729). Although the title “extended physical health status” refers more to physical status rather than qol and sleep, these measures are part of health status in general and a direct consequence of physical health status (lines: 87-98). It would not be "extended" if all these measures were not used together. We also wanted to explore a possible relationship between spinal curvatures angles and quality of life and sleep, which we have added in the Supplementary Table 1.

Reviewer:

The sample size calculation is not clear. What was the actual sample size required for this study? 77 or 123? 

Response:

The appropriate sample size for the study was estimated by guiding two methods as we explained in lines: 138-141: 1) previous similar research and 2) general statistical principles.

1) Sample sizes in similar research ranged from 77 to 168.

Here is the range mentioned, not only “77”.

2) The program MedCalc estimated a minimum of 123 subjects

Although the estimated sample size calculated by MedCalc was 123, we ended up with 82 participants who completed all of the wanted measures.

By presenting other similar research in the method under number “1” above, we wanted to emphasize that this final number of 82 is more than satisfactory. Of course, it is always better to have a larger sample, so we enrolled more than 123, i.e., 145.

However, at this time it is not possible to include more young adults as the study would need to be extended over several years, not just one academic year, in order to have a larger sample with all measures completed.

Reviewer 2 Report

The manuscript has been significantly improved..

Author Response

Reviewer 2:

Comments and Suggestions for Authors

The manuscript has been significantly improved.

Response:

Dear Reviewer,

we would like to thank you for all your comments and the effort to improve our manuscript. It has been a great pleasure to cooperate with you.
